# The Detection of Implanted Radioactive Seeds On Ultrasound Images Using Convolution Neural Networks

**Edward J. Holupka, Ph.D., John Rossman, M.S., Tye Morancy, M.S.,**
Joseph Aronovitz, M.D., Ph.D., Irving D. Kaplan, M.D.

Beth Israel Deaconess Medical Center

Department of Radiation Therapy

Harvard Medical School Boston MA 02215

## Abstract

Purpose: A common modality for the treatment of early stage prostate cancer is the implantation of radioactive seeds directly into the prostate. The radioactive seeds are positioned inside the prostate to achieve optimal radiation dose coverage to the prostate. These radioactive seeds are positioned inside the prostate using Transrectal ultrasound imaging. Once all of the planned seeds have been implanted, two dimensional transaxial transrectal ultrasound images separated by 2 mm are obtained through out the prostate, beginning at the base of the prostate up to and including the apex. Methods: A common deep neural network, called DetectNet was trained to automatically determine the position of the implanted radioactive seeds within the prostate under ultrasound imaging. Results: The results of the training using 950 training ultrasound images and 90 validation ultrasound images. The commonly used metrics for successful training were used to evaluate the efficacy and accuracy of the trained deep neural network and resulted in an loss_bbox (train) = 0.00, loss_coverage (train) = 1.89e-8, loss_bbox (validation) = 11.84, loss_coverage (validation) = 9.70, mAP (validation) = 66.87%, precision (validation) = 81.07%, and a recall (validation) = 82.29%, where train and validation refers to the training image set and validation refers to the validation training set. On the hardware platform used, the training expended 12.8 seconds per epoch. The network was trained for over 10,000 epochs. In addition the seed locations as determined by the Deep Neural Network were compared to the seed locations as determined by a commercial software based on a one to three months after implant CT. The Deep Learning approach was within 2.29 mm of the seed locations determined by the commercial software. Conclusions: The Deep Learning approach to the determination of radioactive seed locations is robust, accurate, and fast and well within spatial agreement with the gold standard of CT determined seed coordinates.

## 1 Introduction

The treatment of early stage prostate cancer using the implantation of radioactive seeds has a long and successful history [1-4]. The typical procedure begins with the acquisition of transaxial ultrasound images of the prostate. Once obtained, computer software is used to determine the optimal placement of the seeds within the prostate. Isotopes that are commonly used for prostate seed implantation are $I^{125}$, $Pd^{103}$, and $Au^{198}$ . The seeds are implanted transperineally into the prostate

1st Conference on Medical Imaging with Deep Learning (MIDL 2018), Amsterdam, The Netherlands.

using 18 gauge needles that are loaded according to the optimal seed locations as determined by the commercial software. Needles are guided into the prostate using a rectangular grid of guide holes. There are 14 guide holes across and 14 guide holes down all equally spaced at 2.5 mm. The needles can be loaded with radioactive seeds as well as plastic spacers. The spacers are the same length as the seeds, which is approximately 0.5 cm. Therefore the planned spatial coordinates of the implanted seeds are discrete. If a coordinate system is chosen such that the x-coordinate goes left to right, the y-coordinate up and down and the z-coordinate into the prostate, the only available seed coordinates are $(i \cdot dx, j \cdot dy, k \cdot dz)$ where $dx = dy = 0.25$ cm and $dz = 0.2$ cm and $i, j, k \in \mathbb{Z}$ are integers.

Once an optimal plan is generated, termed the pre-plan, the seeds are implanted into the prostate according to this treatment plan. This procedure in which the pre-plan is done with in the operating room is termed intra-operative pre-planning. Due to variations in tissue density, elasticity, and operator expertise, the seeds are not deposited exactly as intended. Holupka et. al. has measured this variation and the actual deposited position of the seeds differs on average from the intended position of the seeds by approximately 2 mm [5].

In the current procedure, the patient is released after the implantation and returns for a follow up CT scan approximately six weeks later. A transaxial CT scan of the prostate is obtained at 2 mm image separation. The spatial position of the implanted seeds are then automatically determined using a number of commercially available software applications. In this study Variseed (Varian Medical Solutions, Palo Alto CA) was used. The radiation dose distribution is then determined based on the seed positions as determined by Variseed. This radiation plan can then be evaluated in terms of its compliance to the physicians clinical objective. The clinical objective is defined as the physician's required radiation dose coverage of the Planned Target Volume (PTV) as well as radiation dose constraints on healthy tissue structures such as the urethra, rectum and bladder. This radiation plan is termed the "post-plan". It would be ideal if the post-plan could be performed in the operating room immediately after the implantation. The physician can inform the patient immediately how well the procedure went as well as add additional seeds if a particular area of the prostate received less than the intended dose. Since ultrasound is the modality of choice in the operating room, an automatic post-plan must be based on ultrasound not CT. While many attempts have been made to create an ultrasound based post-planning application, either they have failed or have limited success. This failure is in part due to the difficulty in identifying seeds on ultrasound. Even the "expert" defined seed locations used in this study can arguably be called somewhat accurate at best. In all reality the neural network may in fact be more accurate than the human defined seed positions, which puts the ground truth somewhat in doubt. However, suffice it to say that the real measure of the inferred seed positions as determined by the neural network has to be compared to the "gold standard" of the CT based seed positions. The true accuracy of the neural network's ability to accurately determine seed positions on ultrasound will be measured in this study.

Deep Neural Networks, or Deep Learning has an long and successful history[6-30]. Deep learning has been applied to such applications in medicine as computer assisted pathology readings of digital pathology reports, computer assisted determination of skin legions, and many other medical applications, It was only with the development of the DIGITS and CAFFE platforms pioneered by the Berkeley Vision and Learning Center together with the NVIDIA Corporation (NVIDIA Corporation, 2788 San Tomas Expressway, Santa Clara, CA 95051), which takes advantage of GPU driven escalation of computations needed for the rapid learning seen in these applications. GPUs tend to have many thousand more computational cores than a conventional CPU and have the ability to rapidly teach these deep neural networks to "learn" such things as image recognition, object detection and segmentation on medical images. These platforms are a combination of Python and c++ programs that have been vetted and perform quite well. All of these routines are made transparent by the use of DIGITS, made popular by NVIDIA, which is an HTML based wrapper around these routines. DIGITS makes accessible the use of Deep Learning to the average medical researcher yielding fast and accurate results. NVIDIA currently supplies a range of GPU based graphics cards that can be used in Deep Learning.

CT imaging is currently the gold standard for the determination of radioactive seed locations within the prostate for post intraoperative implantation. However, CT imaging is expensive and not readily available in the operating room. The ability to detect the spatial location of radioactive seeds implanted inside the prostate on ultrasound imaging has been a long standing problem. The solution

should be fast, obtained within a few seconds or minutes, and accurate relative to the spatial location of the seeds as determined by CT imaging.

## 2   Material and Methods

Figure 1 displays a representative sample of transaxial ultrasound images of the prostate after implantation of the radioactive seeds during the intra-operative procedure. The deep neural network, DetectNet (which has a fixed topological structure and commonly used network), was trained to detect implanted radioactive seeds in the prostate under ultrasound imaging. The details of DetectNet, such as the network topology, can be found elsewhere. Some of the parameters of DetectNet were altered to obtain optimal results . DetectNet is a large convolution deep neural network which contains convolution, max pooling, and ReLU (Rectified Linear Units) layers. The training of DetectNet utilized 950 training ultrasound images (which were obtained after the implant so that the seeds were detectable on ultrasound) and validated against 50 ultrasound images. Each ultrasound image was labeled "1.jpg, 2.jpg, ..." with a corresponding text file, "1.txt, 2.txt, ..." which described in the DetectNet format, where the radioactive seeds were located on the ultrasound images. The radioactive seed locations determined in these text files were defined by two of the authors who were determined to be "experts" at defining the seed locations on ultrasound. These locations which surrounded the radioactive seeds were called "bounding boxes", or "bboxes" for short. Two authors met to agree on the seed locations as determined on the ultrasound Images. Once these seed artifacts were located on each ultrasound image, the corresponding text file telling DetectNet where the seeds were located on the images were created. Some of the images contained no seeds to facilitate learning of the neural network in determining seed locations on the ultrasound images. These predetermined bounding boxes constitute the "ground truth" by which the back propagation algorithm was used to determine the best set of neural network weights that would minimize the coverage values. The structure of DetectNet for both the training and validation procedures is well documented[31]. There exist many different back propagation algorithms, each a variant of the gradient decent method. The back propagation used here was the Adaptive Moment Estimation (ADAM) method.

The DetectNet neural network is based on a small grid that covers each image. There are certain metrics which are evaluated during the training and after correction of the weight space with back propagation on all training images. This evaluation of the metrics and corresponding correction of the neural weights are called an "epoch". In this study all 950 ultrasound images in the training set were evaluated. After each epoch, the neural net tries to determine the bounding boxes present on the validation set and compares these predicted bounding box locations to the author specified ground truth bounding boxes found in the corresponding text files. It should be noted that the Deep Neural Network has never "seen" the images in the validation set. It just tries to determine where, if any, the seeds are located on these new validation images. The metrics for the validation set are then determined to evaluate how well the neural network is doing in predicting where the bounding boxes, or radioactive seeds, are located on the validation set. The metrics used are defined here.

Assume an image , $I_\mu$ , $1 \leq \mu \leq N_{images}$ , where $N_{images}$ is the total number of images, $N_{images} = N_{traiing} + N_{validation}$ where $N_{training}$ are the number of training images and $N_{validation}$ is the number of validation images. Let every image be covered by a uniform rectangular grid where each grid rectangle is uniformly $N_{grid}$x$M_{grid}$ pixels.

For convenience define ,

$$N_{TBBI_\mu} = \text{True Number Bounding Boxes per Image } \mu, \tag{1}$$

$$N_{PBBI_\mu} = \text{Predicted Number Bounding Boxes per Image } \mu, \tag{2}$$

$$N_{G_\mu} = \text{Number Grid Rectangles per Image } \mu. \tag{3}$$

Therefore the coverage of a true bounding box of area, $A_{i_\mu}$ , $1 \leq i_\mu \leq N_{TBBI_\mu}$ , $1 \leq \mu \leq N_{images}$ by a grid square $G_{j_\mu}$ is ,

$$coverage\left(A_{i_\mu}\right) = \sum_{J_\mu=1}^{N_{G_\mu}} \frac{Area\left(A_{i_\mu} \cap G_{j_\mu}\right)}{Area\left(G_{j_\mu}\right)} \in [0,1], \tag{4}$$

and the coverage of a predicted bounding box , $B_{j_\mu}$ , $1 \leq j_\mu \leq PBBI_\mu$ , $1 \leq \mu \leq N_{images}$ is ,

$$coverage\left(B_{i_\mu}\right) = \sum_{j_\mu=1}^{N_{G_\mu}} \frac{Area\left(B_{i_\mu} \cap G_{j_\mu}\right)}{Area\left(G_{j_\mu}\right)} \in [0,1] \,.$$ (5)

If

$$IOU_{A_{i_\mu}B_{j_\mu}} = \frac{Area\left(A_{i_\mu} \cap B_{j_\mu}\right)}{Area\left(A_{i_\mu}\right) + Area\left(B_{j_\mu}\right) - Area\left(A_{i_\mu} \cap B_{j_\mu}\right)} > \lambda,$$ (6)

where $\lambda$ is some threshold then $B_{i_\mu}$ is a true positive else it is not a true positive (either true negative or false positive).

The precision and recall for the validation set is defined ,

$$Precision_{validation} = \sum_{\mu=1}^{N_{validation}} \frac{N_{True\,Positives_\mu}}{N_{True\,Positives_\mu} + N_{False\,Positives_\mu}},$$ (7)

$$Recall_{validation} = \sum_{\mu=1}^{N_{validation}} \frac{N_{True\,Positives_\mu}}{N_{True\,Positives_\mu} + N_{False\,Negatives}},$$ (8)

where $N_{True\,Positives_\mu}$ is defined as the number of true positives on image $\mu$ , $N_{False\,Positives_\mu}$ is defined as the number of false positives on image $\mu$ and $N_{False\,Negatives}$ is the number of false negatives on image $\mu$. The $CoverageLoss$ is defined as the $coverage(A) - coverage(B)$. The $mAP$ is the product of the $precision$ and the $recall$, specifically,

$$mAP = \sum_{\mu=1}^{N_{validation}} \frac{N_{True\,Positives_\mu}}{N_{True\,Positives_\mu} + N_{False\,Positives_\mu}} \cdot \sum_{\mu=1}^{N_{validation}} \frac{N_{True\,Positives_\mu}}{N_{True\,Positives_\mu} + N_{False\,Negatives}} \,.$$ (9)

$BboxLoss$ is defined as the differences in the left lower corner of the true bounding box and the predicted right upper corner and left lower corner of the predicted bounding box. Therefore the performance of DetectNet is based on the values of $LossBbox(training)$, $LossBbox\,(validation)$ $LossCoverage\,(valuation)$ ,$mAP\,(validation)$ ,$Precision\,(validation)$ ,
$Recall(valuation)$ and,
$LossCoverage(training)$. The deep neural network adapts the weights per node so as to minimize the valuation loss coverage and the weights are adjusted accordingly. The weights are adjusted according to the gradient descent method used here (ADAM).

## 3   Results

Training DetectNet on the radioactive seed implanted ultrasound images consisted of 950 training images, and 90 validation images. The hardware platform used was a Whisper Station consisting of four Intel Xeon E5-1600v4 CPUs, NVIDIA Quadro K420, 4 NVIDIA GeForce GTX Titan-X GPUs. CUDA and DIGITS came pre-installed using the Ubuntu 16.04 operating system. It took approximately 12.8 seconds per epoch at the current training and validation image set size.

Figure 2 displays a typical learning curve for DetectNet where the loss drops initially and the $mAP$ raises above zero. DetectNet was successfully trained to determine the location of implanted radioactive seeds on ultrasound imaging. Figure 1 also shows some typical inferences from the validation image set. It is noticed that even with a modest $mAP$ value of 66.87% the inferences of where the seeds are located on the validation images is quite exact. The trained DetectNet inference takes approximately 0.2052 seconds per ultrasound image.

Comparison to CT Images: The seed locations of ten patients of the training set images as well as the validation set images were compared to the seed locations as determined by the commercial software called Variseed by Varian Medical Systems, Palo Alto, Ca. Variseed uses a robust algorithm to determine the seed locations on the CT images obtained for each of the patients in this study. These seed locations can be exported as text files and are easily compared to the inferred seed locations determined by trained DetectNet. Since it is not uncommon for the CT images to be obtained one to three months or so after the implant (while the ultrasound images presented to DetectNet are

obtained on the day of the implant procedure) the seed locations may be different by a scale factor since adema may have set in during the procedure. This scale factor is represented asymmetrically as $(S_x S_y, S_z) \in \Re^3$ and represents by how much the prostate gland has spatially changed its shape. In addition the seed locations as determined by DetectNet may be different both translationally and rotationally from that of the CT coordinates. To bring the inferred seed coordinates as determined by DetectNet to be in the same spatial coordinate system as used by Variseed and to determine the minimal distance these seeds can be, a program was developed in the c# programming language, that rotates, translates, and scales the inferred ultrasound seed coordinates, or the coordinates of the seeds as determined by DetectNet, so the minimal distance between the CT coordinates and the ultrasound coordinates are at a minimum. First the program determines the center of mass of all seed artifacts in the ten patients studied and brings both the CT defined seed coordinates to its center of mass as well as the inferred, by DetectNet, ultrasound seed coordinates to their center of mass. This program loops over all rotations and scales in the range of

$$R_{x,y,z} \in \left[-10^o, 10^0\right], \; S_{x,y,z} \in [0.70, 1.30], \tag{10}$$

where $R_{x,y,z}$ is the angular rotation in degrees (ultimately represented as the Euler matrix) in the $x, y,$ or $z$ directions, and $S_{x,y,z}$ is the anisotropic scaling factor (representing adema) in the $x, y,$ and $z$ directions. These transformations are determined for all DetectNet inferred ultrasound seed coordinates as compared to each single, individual seed coordinate of the CT set. The minimal distance obtained, as well as the transformation that determined this minimal distance is reported by the program. While this represents the best of all possible worlds, it is the best minimal distance these seeds can be located relative to each other. The program loops over all possible transformations as determined by the intervals of equation 10.

The results over all seed locations for the seed locations determined by inference of the best DetectNet metric values for the validation set as compared to the CT coordinates as determined by the Variseed software were determined to be within on average 2.29 mm of each other. This reported value is a few millimeters and represents excellent agreement between the trained DetectNet inferred seed locations and the CT based seed locations.

## 4   Discussion

An artificial deep convolution neural network (DetectNet) was trained on thousands of ultrasound images containing implanted seed artifacts. The seeds were trained by a experts (two of the authors) to indicate where the radioactive seeds were located on each image. Approximately 15% of these ultrasound images were used as an example set, termed the validation set, of which the deep convolution network never saw before, and the seeds were inferred on this set as well. The inferred seed locations were compared to the gold standard laid down by the industry of the CT seed coordinates and the inferred seed locations were in fact very close to the CT seed coordinates. 10 patients were studied in this manner and the inferred seed locations compared to the CT seed locations were 2.29 mm of each other.

The training performed on the hardware and software described in this article took a reasonable amount of time using the standard implementation of DIGITS on the UNBUNTU operating system. The weights of the final trained network for each hidden layer (of different types) was stored so that these weights can be used in the forward direction of the network to infer seed locations in the operating room.

## 5   Conclusion

An automated method to detect the spatial positions of radioactive seeds implanted into the human prostate was developed by training the Deep Neural network DetectNet. $LossBbox(training) = 0.00$ , $LossCoverage(training) = 1.89e - 8$, $LossBbox(validation) = 11.84$, $LossCoverage(validation) = 9.70$, $mAP(validation) = 66.87\%$, $Precision(validation) = 81.07\%$, and a
$Recall(validation) = 82.29\%$, where training and validation refers to the training image set and validation refers to the validation training set. The network was trained for over 10,000 epochs. The trained DetectNet inference takes approximately 0.2052 seconds per ultrasound image. There are

typically 8-12 intraoperative ultrasound images taken. This results in approximately 2.052 seconds per patient study.

DetectNet performed very well for the problem of automatically detecting the spatial position on ultrasound imaging of radioactive seeds implanted in the prostate. Future plans for the clinical implementation of DetectNet will be to detect the radioactive seeds and then perform an accurate dose calculation to evaluate the efficacy of the therapy in the operating room.

The agreement with the seed coordinates as determined by the commercial software, Variseed, and the inference seed positions from the CNN was determined to be with 2.29 mm of each other. This type of agreement is excellent for automatic determination of seed locations using either imaging modality, ultrasound or CT[5].

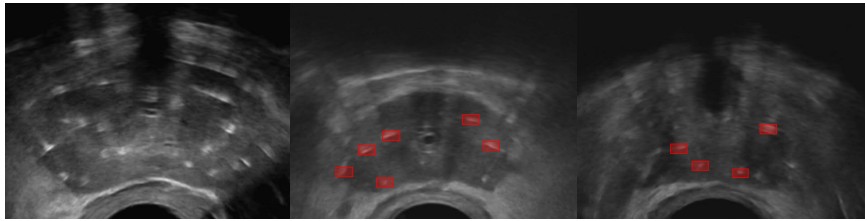

Figure 1: (a) Sample ultrasound images of the prostate, (b,c) inference of where the predicted seeds are on validation images.

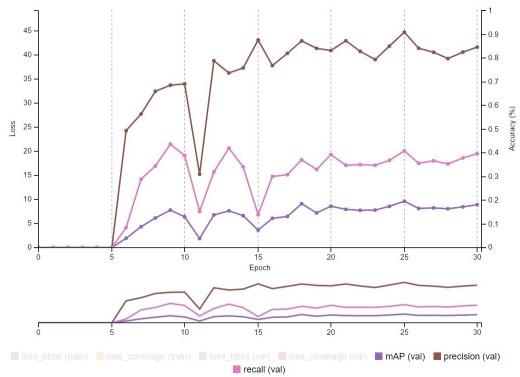

Figure 2: Typical learning curves associated with the training of DetectNet for the purpose of identifying radiation seeds implanted into the prostate.

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
