# OpenReview forum: "The Detection of Implanted Radioactive Seeds On Ultrasound Images Using Convolution Neural Networks "
_MIDL.amsterdam/2018/Conference — Submitted to MIDL 2018_

### Review · AnonReviewer2 · 2018-05-08
**This paper proposes a CNN-based method (called DetectNet) for the automatic determination of the optimal spatial positions of implanted radioactive seeds using prostate ultrasound images. The proposed method yielded consistent performance compared with the gold standard using CT images.**

**Rating:** 2
**Confidence:** 3

**Review:**

Overall, the technical contribution of this paper is limited, considering it presents a direct application of existing deep neural networks in a specific research field (i.e., automatic determination of radioactive seed locations).

Some critical details should be more clearly described. For example, the authors introduced that the details of DetectNet, such as the network topology, can be found elsewhere. However, as the main part of the proposed method, the network architecture and the motivation to choose this kind of network should be indicated.

In the experimental section, the authors only show the results on training and validation sets. What’s the performance on testing set? How can we evaluate the effectiveness of the proposed method?

The quality of figures presented in this paper could be significantly improved. For example, Figure 1 and Figure 2 are not clear.

**Special Issue:**

No

---

### Review · AnonReviewer3 · 2018-05-09
**Interesting and important application of an existing network solution**

**Rating:** 3
**Confidence:** 2

**Review:**

Authors used DetectNet by Nvidia which is an object detection network to find radio-active seeds in ultrasound (US) images, the implantation of which provide optimal radiation dose coverage as a treatment for early stage prostate cancer. Currently, the detection of the seeds is done using CT in post-procedure followups. In order to achieve seed verifications within the operating room (prior to patient discharge)– only US can be used. In this work, an automated network based solution to detecting seeds in US images is proposed and is validated using true CT data. Authors claim that the results are good enough for clinical application shortening procedure times significantly. This is an application of a known Network to an important clinical task.

Strengths-
Immediate clinical use.
Authors who are doctors claim to be satisfied with the results.

Weaknesses -
•	No new methodology developed.
•	The paper is difficult to follow without looking at detectNet representation.
•	mAP measure is not defined in a standard way, here the definition is more similar to F-score;  Can authors explain why precision/ recall are enough for clinical need?
•	In comparing US with CT seeds – a crude registration alg is used and the minimum distances are recorded. Does the number 2.29 mm reflect the average? Of how many seed points?  Should we see the spread of distances? Can we see sample results for the comparisons between US and CT numbers?

Clarity -
•	Some well known measurements (precision,recall,IOU) are described while details about bbox and grid representation are omitted. Why is coverage measure important for results?
•	Many details about software implementation which do not contribute.
•	The learning graph doesn't teach anything new.
•	Too many refs not related to the work at hand

Significance -
The paper is a straightforward application of DetectNet, but improving what seems to be important clinical need.


**Special Issue:**

No

---

### Review · AnonReviewer1 · 2018-05-10
**missing contribution**

**Rating:** 1
**Confidence:** 3

**Review:**

The authors use a ready provided platform from NVIDIA to train a detection network with ultrasound images of implanted radioactive seeds. I neither see the theoretical nor the experimental contribution of this paper.


**Special Issue:**

No

---

### Decision · Program_Chairs · 2018-05-15
**Paper79 Acceptance Decision**

Reject